# Open Problems within Nonextensive Statistical Mechanics

**DOI:** 10.3390/e26020118

**Published:** 2024-01-29

**Authors:** Kenric P. Nelson

**Affiliations:** Photrek, LCC, Watertown, MA 02472, USA; kenric.nelson@photrek.io or kenric.nelson@gmail.com

**Keywords:** complexity, nonextensive, Pareto, student’s t, Fourier, entropy

## Abstract

Nonextensive statistical mechanics has developed into an important framework for modeling the thermodynamics of complex systems and the information of complex signals. To mark the 80th birthday of the field’s founder, Constantino Tsallis, a review of open problems that can stimulate future research is provided. Over the thirty-year development of NSM, a variety of criticisms have been published ranging from questions about the justification for generalizing the entropy function to the interpretation of the generalizing parameter *q*. While these criticisms have been addressed in the past and the breadth of applications has demonstrated the utility of the NSM methodologies, this review provides insights into how the field can continue to improve the understanding and application of complex system models. The review starts by grounding *q*-statistics within scale-shape distributions and then frames a series of open problems for investigation. The open problems include using the degrees of freedom to quantify the difference between entropy and its generalization, clarifying the physical interpretation of the parameter *q*, improving the definition of the generalized product using multidimensional analysis, defining a generalized Fourier transform applicable to signal processing applications, and re-examining the normalization of nonextensive entropy. This review concludes with a proposal that the shape parameter is a candidate for defining the statistical complexity of a system.

## 1. Introduction

Nonextensive statistical mechanics (NSM) [1,2,3] has developed into an important framework for modeling the thermodynamics of complex systems [4,5,6] and the information of complex signals [7,8,9]. The methodology ties together heavy-tailed statistics derived from a generalized entropy function and the resultant analysis, modeling, and design methods for systems impacted by nonlinear dynamics. To mark the 80th birthday of the field’s founder, Constantino Tsallis, I reflect on open problems that will stimulate future investigation and development of NSM. While there is much to celebrate in the applications of NSM, a review of open problems requires examination of some of the criticism [10,11,12,13,14,15] the field has received over its thirty-year development [16]. The criticism ranges from questions about the interpretation of the generalizing parameter *q* to the justification for modifying the entropy function. In this paper, I will carefully examine several key concerns with the aim of motivating the further improvement and applicability of NSM.

Inspection of a few applications of NSM introduces the challenges of characterizing the properties of complex systems using *q*-statistics. Table 1 lists seven examples in which a theoretical underpinning is available to explain experimental observations. However, in each case, the mapping between the physical phenomena and the parameter *q* requires unexplained constants that detract from the ability of NSM to describe the physics of those systems. The relationship q=1+x, in which *x* (or its inverse) is a physical property, is common since *x* often defines a property of the system that induces nonlinearity. This property changes to zero (or infinity for inverse) when q=1. Another typical relationship is q=2−y, since this is the reflection of q=1. If NSM was defined in terms of the physical property *x*, the reflection of 0 would simply be x=−y. However, the full review of open problems in NSM will show that this simple translation is not adequate to account for multidimensional systems and the effects of other nonlinear elements.

The analysis in this review is grounded in the role heavy-tailed statistics plays in modeling the nonlinear dynamics of complex systems. It will be shown that by decomposing the NSM parameter *q* into more direct physical properties, interpretations of NSM are clarified and the connections with the tail shape of distributions, such as the generalized Pareto distribution and the Student t distribution, are simplified. This approach has been called *nonlinear statistical coupling* (NSC). Here, I refer to the theory as NSM and reserve NSC or simply the coupling for the shape parameter, which may also be a candidate for quantifying statistical complexity. For simplification, I will assume that distributions have a location of zero throughout. Also not included in the discussion are distributions, such as the Weibull distribution, which introduce modifications to the skew of the distribution. 

Each section addresses a fundamental question and defines an open problem. In some cases, a comment will be provided suggesting directions for investigation. Solutions are specifically not provided because although the author has, in some cases, previously recommended a solution, the future direction of NSM is ultimately a community decision made by the investigators pushing the field forward. Section 2 reviews how *q* relates to the traditional parameters of heavy-tailed and compact-support distributions. Section 3 discusses the question of generalizing entropy. Section 4 examines the difference between mathematical fits and physical theories, as well as the role of independent random variables in clarifying the physical property of *q*. Section 5 highlights some inconsistencies in how the *q*-product is defined and applied. Section 6 explains the limitations in the use of the *q*-Fourier transform as a physical model. Section 7 considers three different normalizations of the nonextensive entropy. Finally, Section 8 asks whether a definition for statistical complexity is possible. 

## 2. How Is *q*-Statistics Related to Traditional Definitions of Heavy-Tailed Distributions?

NSM began [16,26] with a proposal to generalize Boltzmann–Gibbs statistics by examining the properties of systems with a distribution of states modified by the power of *q*. This modified distribution with the elements $p_{i}^{q}$ is referred to as the escort distribution; however, it is unfortunate that the NSM literature has not been explicit that this expression necessarily defines *q* as a real number of independent random variables sharing the same state. If *q* is an integer *n*, elementary probability theory establishes that $p^{n}$ is the probability of n independent random variables, each with probability p. Fractional random variables are discussed further in Section 4. From this start, the Tsallis entropy and its maximizing distribution were derived as follows:(1)$$S_{q}^{T}\equiv \frac{1-\sum_{i}p_{i}^{q}}{q-1}$$
(2)$$p_{i}=\frac{{\left(1-\beta \left(q-1\right)x_{i}\right)}^{\frac{1}{q-1}}}{\sum_{j=1}^{N}{\left(1-\beta \left(q-1\right)x_{j}\right)}^{\frac{1}{q-1}}}$$

**Warm-Up Problem:** *The first problem is not so much open as a warm-up to ground the discussion of the other problems. How does the NSM parameter* q *relate to the shape of a distribution? And how does the Lagrange multiplier* β *relate to the scale of a distribution?*

I will address this question via the examination of the generalized Pareto and Student t distributions. Both the probability density function (pdf) and the survival function (sf) are provided since the sf will provide insights into the definition of a generalized exponential function. To unify the discussion, both distributions will be defined in terms of the shape parameter κ, though the Student t is traditionally defined in terms of its reciprocal, the degrees of freedom, ν=1κ. The shape parameter is also referred to as the nonlinear statistical coupling or coupling due to its connection with nonlinearity; further, the final problem will consider whether it is a candidate for quantifying statistical complexity. The distributions have three domains:(3)$$\begin{array}{cc} Compact-Support & -1<\kappa <0\\ Exponential & \kappa =0\\ Heavy-Tail & \kappa >0 \end{array}$$

**Definition 1:** 
*Generalized Pareto Distribution*


*The survival function (cf) is one minus the cumulative distribution function (cdf), *$\bar{F}=1-F$*. The Pareto Type IV with a location of zero is defined in terms of a scale,*σ*, and two shape parameters,*κ *and* α.(4)$$\bar{F}(x;\sigma ,\;\kappa ,\alpha )={\left[1+\kappa {\left(\frac{x}{\sigma }\right)}^{\alpha }\right]}^{-\frac{1}{\alpha \kappa }};x\geq 0,\;\kappa ,\;\alpha >0$$


*The probability distribution function (pdf) is the derivative of the cdf*

(5)
$$\left(x;\sigma ,\;\kappa ,\alpha \right)=\frac{1}{\sigma }{\left(\frac{x}{\sigma }\right)}^{\alpha -1}{\left[1+\kappa {\left(\frac{x}{\sigma }\right)}^{\alpha }\right]}^{-\left(\frac{1}{\alpha \kappa }+1\right)};x\geq 0,\;\kappa ,\;\alpha >0$$



*For Pareto Type II* α=1*and the cf and pdf reduce to*(6)$$\bar{F}\left(x;\sigma ,\;\kappa \right)={\left[1+\kappa \frac{x}{\sigma }\right]}^{-\frac{1}{\kappa }};x\geq 0,\;\kappa >0$$(7)$$f\left(x;\sigma ,\;\kappa \right)=\frac{1}{\sigma }{\left[1+\kappa \frac{x}{\sigma }\right]}^{-\left(\frac{1}{\kappa }+1\right)};x\geq 0,\;\kappa >0$$

**Comment on Definition 1:** *The definition for Type IV is modified from the traditional approach to clearly distinguish between the decay of the tail in the limit as* x *changes to infinity,* κ*, and the raising of the variable to the power* α*. Thus, the outer exponent is* $-\frac{1}{\alpha \kappa }$*, meaning that* $-\frac{1}{\kappa }$ *is the asymptotic power. Nevertheless, the emphasis here will be on questions about NSM and connections to the long-standing traditions in statistical analysis.*

**Definition 2:** 
*Generalized Student t Distribution*


*The Student t distribution is traditionally defined in terms of the degrees of freedom,* ν*; however, to unify the discussion, the reciprocal shape parameter,* κ=1ν*, is used. The survival function of the generalized Student t distribution, which depends on the Gauss hypergeometric function, _2_F_1_, and the Beta function, B, is*(8)F¯x;σ,κ,α=−12−σκB12,  κ−signκκ−24κF1212, 1+κ2κ; 32;min1,−κx2σ2; 0≤x≤−κ−1<κ<00≤x<∞κ≥0(9)F12a,b;c;z=∑n=0∞anbncnznn!; an=1n=0aa+1⋯a+n−1n>0;Bz1,z2=∫01tz1−1t−1z2−1dt.*The Student t probability density function is:*(10)$$f\left(x;\sigma ,\;\kappa \right)=\left\{\begin{array}{cc} \frac{\sqrt{\left|\kappa \right|}}{\sigma B\left(\frac{1}{2},\;\;\frac{\kappa -sign\left(\kappa \right)\left(\kappa -2\right)}{4\kappa }\right)}{\left[1+\kappa {\left(\frac{x}{\sigma }\right)}^{2}\right]}_{+}^{-\frac{1}{2}\left(\frac{1}{\kappa }+1\right)} & \kappa \neq 0,\;\kappa \geq -1\\ \frac{1}{\sigma \sqrt{2\pi }}exp\left(-\frac{1}{\alpha }{\left(\frac{x}{\sigma }\right)}^{2}\right) & \kappa =0. \end{array}\right.$$

**Warm-Up Solution:** *The exponent of the Pareto* $\left(\alpha =1\right)$ *and Student t* $\left(\alpha =2\right)$ *distributions determines the relationship between q and the shape* κ*:*



(11)
$$q=1+\frac{\alpha \kappa }{1+\kappa };\;\kappa =\frac{q-1}{\alpha +1-q}.$$

*From this relationship, the escort probability or density can be defined in terms of the following shape:*

(12)
$$p_{i}^{\left(\alpha ,\kappa \right)}\equiv \frac{p_{i}^{1+\frac{\alpha \kappa }{1+\kappa }}}{\sum_{j=1}^{N}p_{j}^{1+\frac{\alpha \kappa }{1+\kappa }}};\;f^{\left(\alpha ,\kappa \right)}\left(x\right)\equiv \frac{f^{1+\frac{\alpha \kappa }{1+\kappa }}\left(x\right)}{\int_{x\in X}f^{1+\frac{\alpha \kappa }{1+\kappa }}\left(x\right)dx}$$

*The multiplicative term of the variable determines the relationship between the Lagrange multiplier *

β

* and the scale *

σ

*as follows:*

(13)
$$\beta =\frac{\left(1+\kappa \right)}{\alpha \sigma^{\alpha }};\;\sigma ={\left(\frac{1}{\left(\alpha +1-q\right)\beta }\right)}^{\frac{1}{\alpha }}.$$



**Open Problem 1:** *We notice that the Pareto Type II survival function is in the form of the generalized exponential function*(14)$${exp}_{\kappa }\left(z\right)\equiv \left\{\begin{array}{cc} {\left(1+\kappa z\right)}_{+}^{\frac{1}{\kappa }} & \kappa \neq 0,\;{\left(a\right)}_{+}\equiv max\left(0,a\right)\\ e^{x} & \kappa =0. \end{array}\right.$$*This leads to a question regarding the definitions for the generalized algebra of NSM. In the development of NSM, the generalization of the exponential function has been applied to the pdf; however, would the sf be the more natural function to generalize? If so, the shape parameter rather than q becomes the fundamental parameter of the NSM generalization of statistical mechanics. We will see that this modification leads to a clearer definition of the multivariate distributions and more direct physical interpretations. A related issue is that *${exp}_{\kappa }\left(-z\right)\neq {exp}_{\kappa }^{-1}\left(z\right)$ *for* κ≠0*. This is important in the definition of distributions since it is the reciprocal of the exponential function rather than the negative of the argument that is important.*


*Before continuing, the inverse of the generalized exponential function is defined as the generalized logarithm as follows:*

(15)
$$\log_{\kappa }z\equiv \left\{\begin{array}{cc} \frac{1}{\kappa }\left(z^{\kappa }-1\right) & \kappa \neq 0,\;z>0\;\\ e^{z} & \kappa =0,\;z>0. \end{array}\right.$$



## 3. Is a Generalization of Entropy Necessary?

One of the challenges of statistical mechanics is that it is quite difficult even for seasoned experts to formulate an intuitive framework for its foundational concept, entropy. To address the question of the need for a generalized entropy, we will describe the issue in terms of a distribution’s average density (or probability for non-continuous distributions). While most concepts in statistics are framed in terms of densities/probabilities (y-axis of distribution) and estimates of the random variable (x-axis of distribution), entropy is based on the logarithm of the probabilities. This transformation, p→log⁡p, is essential to providing an additive scale, meaning that the arithmetic average is the central tendency of the uncertainty, leading to the definition of entropy, $S=-\sum_{i=1}^{N}p_{i}logp_{i}$. This is the informational entropy, which will be used in this paper, while the physical entropy includes multiplication using the Boltzmann constant. We must notice, however, that the logarithm can be separated from the aggregation of the probabilities using the weighted geometric mean $S=\log\prod_{i=1}^{N}p_{i}^{-p_{i}}$. For the continuous distributions, the entropy is $S=-\int_{x\in X}f\left(x\right)\log f\left(x\right)dx$, and the equivalent of the weighted geometric mean of the density is $exp\left(-S\right)$, known as the log-average. Therefore, the weighted geometric mean can be used to examine the *average density or probability* without resorting to the logarithmic transformation. Using Equations (12)–(15), the log-average is generalized to a function that I will refer to as the coupled log-average.
(16)$$f_{avg}\left(x;\alpha ,\kappa \right)\equiv {\left({exp}_{\kappa }\left(\int_{x\in X}f^{\left(\alpha ,\;\kappa \right)}\left(x\right)\log_{\kappa }f^{-\frac{\alpha }{1+\kappa }}\left(x\right)dx\right)\right)}^{-\frac{1+\kappa }{\alpha }},$$
where the factor $-\frac{\alpha }{1+\kappa }$ and its inverse are determined using the exponent of the distribution *f*. For discrete functions, (9) reduces to the generalized mean, as derived from Definition 3 of [27].
(17)$$p_{avg}\left(p;\alpha ,\;\kappa \right)\equiv {\left(\sum_{i=1}^{N}p_{i}^{\left(\alpha ,\;\kappa \right)}p_{i}^{-\frac{\alpha \kappa }{1+\kappa }}\right)}^{-\frac{1+\kappa }{\alpha \kappa }}={\left(\sum_{i=1}^{N}p_{i}^{1+\frac{\alpha \kappa }{1+\kappa }}\right)}^{\frac{1+\kappa }{\alpha \kappa }}.$$

Figure 1 shows the Gaussian, κ=0, and three heavy-tailed coupled Gaussians, κ={0.5, 1, 2}. The distributions are normalized by their couple average density, which is highlighted in the figure by a horizontal line. The couped average density is computed for each density with the matching coupling value, *κ*. Furthermore, the matching coupled average of the density is always equal to the density at x=μ±σ. As the coupling or shape increases, the tail becomes heavier, and the log-average $\left(\kappa =0\right)$, shown as dashed horizontal lines, approaches zero. Thus, the entropy, which is the logarithm of the average density, approaches infinity. Nevertheless, the Student t distribution has a structure that is quite different from the variance of the Gaussian changing to infinity. Something has clearly been lost in summarizing the uncertainty of the Student t with just the entropy.

In particular, the scale *σ* of the Student t distribution, which generalizes the standard deviation of the Gaussian and is referred to as the *q-standard deviation* in NSM, remains finite. The analysis shows that the generalized mean can be used to separate the effect that the shape and the scale have on measures of the uncertainty. 

**Open Problem 2:** 
*Given that the average generalized density is equal to the density at the mean plus/minus the scale for the coupled Gaussian and the location plus the scale for the coupled exponential, can the relationship between the average generalized density and the average density be quantified in a manner that strengthens the explanation of how the generalized entropy complements the entropy function in describing the uncertainty of a system? For instance, given that entropy is a measure of the degrees of freedom of a system, and the coupling is the inverse of the degrees of freedom, can the difference between the coupled entropy and the entropy be quantified in terms of the degrees of freedom?*


**Comment on Problem 2:** *An important aspect of the investigation of statistical degrees of freedom is its relationship with the thermodynamic degrees of freedom. As noted in Table 1 and described in* [25], *q is determined by the degrees of freedom, n, of a temperature bath. Substituting (11), the shape, which is the reciprocal of the statistical degrees of freedom, is related by*


(18)
$$n=\frac{d\;N}{2}=\frac{q}{q-1}=2+\frac{1}{\kappa }\;,$$


*Taking* α=1*, given that the distribution is based on the energy. d is the dimensions of translational degrees of freedom, though rotational and vibrational could also be considered. N is the number of molecules.*

## 4. NSM: Mathematical Fit or Physical Theory?

A common criticism of the NSM has been that it is merely a mathematical fit to physical phenomena given a free parameter rather than a physical theory that provides an explanatory description of complex systems [10,12,14,28]. While this claim has been refuted by investigators in the NSM community [2,29,30,31], we should take a moment to consider what distinguishes a physical theory from a mathematical fit. Firstly, mathematical theories build from assumed axioms and deductively prove derivative theories. Physical theories are a subset of mathematical theories that are constrained by physical measurements of the world. So, demonstrating a fit between a mathematical theory and physical measurement is a crucial step toward a physical theory. But is a fit sufficient to qualify a relationship as a physical theory? In physics, we are seeking models that provide explanatory power in describing a system. As such, each term (variables and constants) in a physical theory must have a clear definition of its role in the physical model; otherwise, the model loses its ability to be explanatory.

Further, an effective model must fulfill the requirement of being the simplest representation of a phenomenon. Occam’s razor [32,33] was one of the first articulations of this principle, and Bayes’ Theorem quantifies this property by specifying the uncertainty created by the overfitting of more complex models (see Ch. 28 of [34]). In the context of NSM, these criteria establish a requirement that its defining parameter *q* has a clear physical definition and this property provides a simpler explanation of the statistics of complex systems than the shape or degrees of freedom parameters that it seeks to replace. For even in the case where the equations of NSM can be derived from first principles [35,36], if the defining parameter does not have a physical definition, the derivation still lacks a physical interpretation. 

Furthermore, as noted in the introduction, *q* does in fact have a straightforward interpretation based on the escort distribution. The original motivation of *q*-statistics was the consideration of systems defined by an escort distribution with probabilities $\frac{p_{i}^{q}}{\sum_{i=1}^{N}p_{i}^{q}}$. The quantity $p_{i}^{q}$ defines the probability of *q* random variables that occupy the same state *i*. Thus, the necessary starting point for defining a physical property of *q* is the number of independent random variables sharing the same state. For continuous random variables, we can consider an approximate threshold to discretize the limit. The relevance to complex systems is that the independent components of a multivariate heavy-tailed distribution are nevertheless correlated (or conversely, if linearly uncorrelated the components are dependent). Due to the nonlinear dependence between the dimensions of a heavy-tailed distribution, there is a higher occurrence of discrete variables that are equal or continuous variables that are approximately equal than would occur for distributions with exponential decay. This property has recently been explored as an approach to filtering heavy-tailed samples to facilitate the estimation of their distribution [37].

Nevertheless, the question remains whether the property of equal-valued independent random variables is central to describing the statistics of complex systems. Several investigators have suggested other interpretations, but close examination shows that the descriptions are equal to q−1 or another function of *q*, rather than *q* itself. For instance, Wilk and Włodarczyk [38,39] show how the fluctuations (relative variance) of an inverse scale parameter $\left(\frac{1}{\lambda }\right)$ are equal to q−1. The problem is this does not provide an interpretation of *q*’s statistical property; rather, it shows that *q* is misaligned by −1 with a possible interpretation. The relative variance is indeed a useful property, and, thus, the variable a=q−1 is a candidate for an approach to defining nonextensive statistical mechanics. But, as we will see in the following section, multidimensional analysis shows that neither *q* nor q−1 are fundamental.

**Open Problem 3:** 
*We must define and provide evidence for a physical definition of the parameter q. Included in this definition must be an explanation of the role of the number of independent random variables via the expression $p^{q}$*



*. We must demonstrate that this physical property simplifies and/or improves the description of the statistics of complex systems in comparison to the shape or the shape’s inverse, the degrees of freedom.*


## 5. The *q*-Product Does Not Construct the Multivariate Distributions

Borges [26] initiated and other investigators [40,41,42] further developed a *q*-algebra to encapsulate the core functions of nonextensive statistical mechanics. The foundational functions are a generalization of addition and multiplication, though the two do not form a generalized distributive property. The lack of distribution property was in part caused by the *q*-sum being primarily relevant to the combining of *q*-logarithms, while the *q*-product was primarily relevant to the combining of *q*-exponentials. Here are the definitions in those contexts:(19)$$\ln_{q}x\;{⨁}_{q}\ln_{q}y\equiv \left(\frac{x^{1-q}-1}{1-q}\right)+\left(\frac{y^{1-q}-1}{1-q}\right)+(1-q)\left(\frac{x^{1-q}-1}{1-q}\right)\left(\frac{y^{1-q}-1}{1-q}\right)\;=\left(\frac{{\left(xy\right)}^{1-q}-1}{1-q}\right)=\ln_{q}xy;$$
(20)$$\begin{array}{ll} {exp}_{q}\left(x\right){⨂}_{q}{exp}_{q}\left(y\right) & \\ & \equiv {\left({\left({\left(1+\left(1-q\right)x\right)}_{+}^{\frac{1}{1-q}}\right)}^{1-q}+{\left({\left(1+\left(1-q\right)y\right)}_{+}^{\frac{1}{1-q}}\right)}^{1-q}-1\right)}^{\frac{1}{1-q}}\\ & ={\left(1+\left(1-q\right)\left(x+y\right)\right)}_{+}^{\frac{1}{1-q}}={exp}_{q}\left(x+y\right). \end{array}$$

While these constructions provide a useful shorthand for some of the complex relationships in nonextensive statistical mechanics, when applied to statistical analysis, a significant shortcoming is evident. A bedrock principle of probability theory, which was discussed in the last section, is that independent probabilities multiply to form the joint probability. Thus, a natural question arises regarding the properties of the *q*-product of probabilities. Putting aside for a moment the normalization of the *q*-exponential and *q*-Gaussian distributions, which add a further complication, how does the *q*-product of their distributions relate to the multivariate forms of these distributions? From the definition of the *q*-product, we have
(21)$${exp}_{q}\left({\frac{1}{\alpha }x_{1}^{\alpha }}_{1}\right){⨂}_{q}{exp}_{q}\left(\frac{1}{\alpha }x_{2}^{\alpha }\right)\ldots {⨂}_{q}{exp}_{q}\left(\frac{1}{\alpha }x_{n}^{\alpha }\right)={exp}_{q}\left(\sum_{i=1}^{n}x_{i}^{\alpha }\right).$$
where α is one for the *q*-exponential distribution and two for the *q*-Gaussian. Unfortunately, the expression on the right has very little to do with the multivariate form of these distributions when no cross-terms $x_{i}^{\alpha }x_{j}^{\alpha }$ exist. This is because the exponents of the distribution include both a dimensional term and α. Even for just the one-dimensional case, this led investigators to define $1-Q\equiv 2\left(1-q\right)$ to account for the distinctions. The multivariate form of these distributions [43] is proportional to the following:(22)$$f\left(x\right)\propto {\left[1+\kappa \frac{\sum_{i=1}^{n}x_{i}^{\alpha }}{\sigma^{\alpha }}\right]}^{-\frac{1}{\alpha }\left(\frac{1}{\kappa }+d\right)}.$$
From this expression, it is evident that trying to force the multiplicative term inside the brackets and the exponent to be 1−q and $\frac{1}{1-q}$, respectively, results in several distortions of the physical properties. Firstly, in NSM, the physical scale of the distributions *σ* is typically buried in a parameter referred to as the generalized inverse temperature, $\beta_{q}=\frac{\kappa }{(1-q)\sigma^{\alpha }}$. And from the exponent, *q* is defined by the relationship $\frac{1}{1-q}=-\frac{1}{\alpha }\left(\frac{1}{\kappa }+d\right)$.
(23)$$q=1+\frac{\alpha \kappa }{1+d\kappa }$$
To address the multivariate case, Umarov and Tsallis [44] formulated the following definitions for the multivariate Gaussian case $\left(\alpha =2\right)$:(24)$$q_{k}^{d}\equiv q_{kd}\equiv \frac{2q-kd\left(q-1\right)}{2-kd\left(q-1\right)}.$$
Far from illuminating the multivariate statistics of complex systems, these types of expressions provide evidence that *q* is not aligned with the statistical properties of complex systems. Again, to interpret such an expression, it is not enough to understand that *q* is the number of equal-valued independent random variables; rather, we also need to explain the physical role of each term in the right-hand expression. Without these explanations, the relationship fulfills a mathematical fit but falls short of a physical theory.

**Open Problem 4:** *While the q-product is often referenced regarding its role in defining q-independence, the form does not lead to the structure of the multivariate heavy-tailed distributions in the manner that the product of distributions equates with the multivariate distribution of independent variables. A definition of the generalized product for NSM is needed that is based on the properties of the multivariate distributions, as was proposed in* [43].

## 6. Does the *q*-Fourier Transform Model the Properties of Complex Signals?

One of the celebrated results of NSM is the proof of a generalized central limit theorem (*q*-CLT) [45] that converges to *q*-Gaussians for random variables found to have a property of *q-independence*. The nonlinear dependence described by *q*-independence relies on a generalization of the Fourier transform that maps *q*-Gaussians to a *q**-Gaussians. Given a more general form of the Fourier transforms, a natural application would be the design of filters for signals with long-range correlations and/or fluctuations, the tell-tale characteristic of signals from a complex system. And yet, to date, there appear to be no applications of the *q*-Fourier transform to signal processing. Related to the lack of applications is the lack of a symmetric inverse [46,47,48], one of the key properties that has made the Fourier transform the foundation of signal processing. To frame this problem, I will examine the Fourier transform in the context of the symmetry between the compact-support and heavy-tailed functions of NSM [49]. 

The gap in what should be a straightforward application of NSM results from the disconnect between the mathematical relationships for the *q*-CLT and the physics of signal processing. We must recall that the Fourier transform takes a function as an input (called the signal) and outputs another function (called the image) that preserves all the information about the original function. The process can be inverted with a function that has the same structural form. The image has been proven to represent the sinusoidal frequencies of the original signal and is used throughout engineering and science to craft filters for noise reduction, match filtering, and countless other purposes. As the name “image” implies, the FT is a kind of mirror. When applied to probability distributions, the FT mirror has the property of transforming wide, high-entropy distributions into narrow, low-entropy image functions. The Gaussian turns out to be the symmetrical function of this process, whereby the FT of a Gaussian is also Gaussian (though no longer normalized to integrate to one). And the variance of the image is proportional to the inverse of the signal’s variance.

Unfortunately, as currently defined, the *q*-FT violates this basic relationship between a signal and its image. The *q*-FT transforms both the tail shape and the scale of the distribution. Focusing on the tail shape, the transformed value of *q* and its translation into the shape parameter are determined from the definition of *q*-FT to be
(25)$$q_{1}=\frac{1+q}{3-q};\;\kappa_{1}=\frac{\kappa }{1-\kappa }.$$

Table 2 shows how different domains of the heavy-tailed *q*-Gaussian distributions are transformed by the *q*-FT into wider-tailed images. The Cauchy distribution (κ=1, q=2) highlights the difficulties involved in applying the *q*-FT to signal processing since the image function is an impulse function (κ=∞, q=3). This suggests that the Cauchy distribution is the limit of physically realizable distributions. There are systems such as the Standard Map in which the Cauchy does act as a limiting distribution [50,51]. At the same time, the coherent noise model [52] and the Erhenfest dog–flea model [53] have been measured to have *q*-Gaussians with the shape/q values (κ=1.53, q=2.21) and (κ=2.08, q=2.35), respectively. And yet, distributions in this domain of very slow tail decay (1<κ<∞, 2<q<3) have a *q*-FT image function with a divergent integral (−∞<κ<−1, 3<q<∞), suggesting that these would not arise in physically realizable systems. 

In contrast to the *q*-FT, the Fourier transform maps heavy-tailed *q*-Gaussians into functions that are a product of a power-law term and a modified second-order Bessel function, which has an exponential tail decay.
(26)F1+κx2+−121κ+1∝tκ12κK12tκ.
The power-law term increases with *x* but is sharply dampened by and in the limit of x→∞ dominated by the exponential decay of the Bessel function. It is noteworthy that the exponent of the power-law term $\frac{1}{2\kappa }$ turns out to be a conjugate mapping between the exponents of the heavy-tailed and compact-support *q*-Gaussians. That is, for κ>0, the heavy-tail and compact-support domains are related by
(27)$$\begin{array}{cc} Heavy-tailed & Compact-support\\ {\left[1+\kappa x^{2}\right]}_{+}^{-\frac{1}{2}\left(\frac{1}{\kappa }+1\right)} & {\left[1-\frac{\kappa }{1+\kappa }x^{2}\right]}_{+}^{\frac{1}{2\kappa }}. \end{array}$$
In [49], I proposed a symmetrical generalization of the Fourier transform that maps the *q*-Gaussians between their compact-support and heavy-tailed domains. However, the transform included a mapping of the *q* parameter that did not generalize to other functions. A requirement for a complete definition is a mathematically rigorous mapping between the infinite domain of the heavy-tailed distributions and the finite-domain compact-support functions. 

**Open Problem 5:** 
*As currently defined, the q-Fourier transform of NSM has limited physical applications, since (a) the inverse is not symmetric and (b) the image function has slower decaying tails. Can these limitations be validated by limits within physical applications of heavy-tailed distributions, or can a symmetric generalization of the Fourier transform be defined? A candidate for a symmetric Fourier transform maps q-Gaussians between their heavy-tailed and compact-support domains but currently lacks a general mapping between these domains. Can a mathematically rigorous mapping between the compact-support and heavy-tailed domains be defined that would qualify as a generalization of the Fourier transform?*


## 7. How Should Nonextensive Entropy Be Normalized?

During the early investigations of nonextensive entropy, a question arose regarding the proper probability required to weight the generalized entropy. Tsallis entropy, $S_{q}^{T}\equiv \frac{1-\sum_{i}p_{i}^{q}}{q-1}=-\sum_{i}p_{i}^{q}{ln}_{q}p_{i}\;(Tsallis,2009)$, is weighted by $p_{i}^{q}$; however, this form does not make use of the escort probability $\frac{p_{i}^{q}}{\sum_{i}p_{i}^{q}}$ normalization used for defining the constraints for the generalized maximum entropy formalism. For this reason, the normalized Tsallis entropy [54,55] $S_{q}^{NT}\equiv \frac{S_{q}^{T}}{\sum_{j}p_{j}^{q}}=\frac{-1+\frac{1}{\sum_{i}p_{i}^{q}}}{q-1}=-\sum_{i}\frac{p_{i}^{q}}{\sum_{j}p_{j}^{q}}{ln}_{q}p_{i}$ was investigated. The normalized Tsallis entropy was found not to satisfy the Lesche stability requirement [56,57,58] and has since been dismissed in favor of the original Tsallis entropy form.

However, given the insight regarding the distinction between the power and normalization for the generalized exponential and logarithms discussed in Section 5, another normalization can be considered. The coupled entropy [27,59] is defined as follows:(28)$$\begin{array}{ll} S_{\kappa }^{C}\left(p;d,\alpha \right) & \equiv \sum_{i}\frac{p_{i}^{1+\frac{\alpha \kappa }{1+d\kappa }}}{\alpha \sum_{j}p_{j}^{1+\frac{\alpha \kappa }{1+d\kappa }}}{ln}_{\kappa }p_{i}^{\frac{-\alpha }{1+d\kappa }}\\ & =\sum_{i}\frac{1}{\alpha \kappa }\frac{p_{i}^{1+\frac{\alpha \kappa }{1+d\kappa }}}{\sum_{j}p_{j}^{1+\frac{\alpha \kappa }{1+d\kappa }}}\left(p_{i}^{\frac{-\alpha \kappa }{1+d\kappa }}-1\right). \end{array}$$

In [27], it was found that the coupled and Tsallis entropy are (constant, asymptotically constant) a function of the coupling for the generalized Pareto distribution with the matching coupling value and a scale equal to (one, non-one), respectively. However, for the matched coupled Gaussian, the coupled entropy rose in value, while the Tsallis entropy decayed with the increasing coupling value, i.e., the more heavy-tailed distributions. 

The issue of normalization for a generalized entropy comes into sharper focus when considering the role of a generalized sum in defining the nonlinear combination of entropies. Substituting for *q* the relationship between the coupled, normalized, and Tsallis entropies results in the following expression:(29)$$S_{\kappa }^{C}\left(p;d,\alpha \right)=\frac{S_{\kappa }^{NT}\left(p;d,\alpha \right)}{1+d\kappa }=\frac{S_{\kappa }^{T}\left(p;d,\alpha \right)}{(1+d\kappa )\sum_{j}p_{j}^{1+\frac{\alpha \kappa }{1+d\kappa }}}.$$
While the *q*-sum of q-entropies has been defined as (using the coupling notation)
(30)$$\begin{array}{l} S_{1+\frac{\alpha \kappa }{1+d\kappa }}^{T}\left(p_{A}\right){⨁}_{1+\frac{\alpha \kappa }{1+d\kappa }}S_{1+\frac{\alpha \kappa }{1+d\kappa }}^{T}\left(p_{B}\right)\\ \equiv S_{1+\frac{\alpha \kappa }{1+d\kappa }}^{T}\left(p_{A}\right)+S_{1+\frac{\alpha \kappa }{1+d\kappa }}^{T}\left(p_{B}\right)\\ -\frac{\alpha \kappa }{1+d\kappa }S_{1+\frac{\alpha \kappa }{1+d\kappa }}^{T}\left(p_{A}\right)S_{1+\frac{\alpha \kappa }{1+d\kappa }}^{T}\left(p_{B}\right), \end{array}$$
the coupled sum removes the dependency on $\frac{1}{1+d\kappa }$
(31)$$S_{\kappa }^{C}\left(p_{A}\right){⨁}_{\alpha \kappa }S_{\kappa }^{C}\left(p_{B}\right)\equiv S_{\kappa }^{C}\left(p_{A}\right)+S_{\kappa }^{C}\left(p_{B}\right){+\alpha \kappa \;S}_{\kappa }^{C}\left(p_{A}\right)S_{\kappa }^{C}\left(p_{B}\right).$$
Notably, the B-G-S entropy scales with the degrees of freedom, and the nonextensive entropies modify this scaling [35,60]. Given that κ is the inverse of the statistical degrees of freedom, the coupled sum of the coupled entropies directly expresses this modification.

**Open Problem 6:** 
*What is the proper normalization of a generalized entropy, and how does the normalization impact the relationship between a generalized entropy and the statistical degrees of freedom? Stability issues caused a rejection of the normalized Tsallis entropy; however, neither the normalized nor the unnormalized Tsallis entropy consider how converting the derivatives of a cdf into a pdf impacts the relationship between the definition of the generalized exponential and logarithmic functions and the structure of a pdf and its generalized entropy. When this is accounted for, the nonlinear term combining a nonextensive entropy (coupled entropy) is precisely the inverse of the statistical degrees of freedom. Does this suggest a criterion for the normalization?*


**Comment on Problem 6:** *There are a variety of applications that may be impacted by the normalization of the NSM entropy. For instance, the robustness of machine learning algorithms have been improved using both q-entropy* [61,62] *and coupled entropy* [59] *generalizations. A careful analysis of whether the difference in normalization impacts the performance improvements would contribute to determining the importance of the normalization. Entropic analysis has been shown to be an effective measure of financial market volatility, but greater detail is needed to determine the relative advantages of different forms of generalized entropy* [63,64]. 

## 8. A Measure of Complexity

The derivation of nonextensive entropy began with the investigation of systems with a modified distribution, in which the probability of a state is raised to the power *q*. As discussed in Section 2 and Section 4, this necessitates a physical interpretation of *q* as the number of independent random variables sharing the same state. Unfortunately, clarifying this interpretation raises questions as to whether *q* is a fundamental or secondary property of complex systems. The more fundamental question is as follows: how should the statistical complexity of a system be quantified? The mismatch between *q’s* physical interpretation and the fundamental properties of complex systems may explain why the field has avoided addressing this issue.

Nevertheless, an approach to quantifying the statistical complexity of a system may be quite simple. The property nonlinear statistical coupling was first introduced with the candidate 1−q, which fulfilled the need for the linear domain to have a value of zero; however, multidimensional analysis exposed that isolating the nonlinear properties required decomposition. As shown in Equation (23), *q* is dependent on three properties, the dimension *d*, the nonlinearity of the random variable α, and the redefined nonlinear statistical coupling κ. The coupling term is not new; in fact, it has a long tradition within statistical analysis as the shape parameter defining the deviation from exponential decay, and it is the inverse of the degrees of freedom used to define the Student t distribution. And so, the final open problem for the reader to consider is whether the coupling or shape parameter is an appropriate measure of a system’s statistical complexity.

**Open Problem 7:** 
*Does the shape parameter, also referred to as the nonlinear statistical coupling, provide a quantification of a system’s statistical complexity? Can this definition of statistical complexity be related to other forms of complexity, such as algorithmic complexity? Explain the statistical complexity in terms of its inverse, the statistical degrees of freedom. For instance, given samples from which to determine a model, does the nonlinearity of the function define the deterministic complexity of the model? And do the statistical degrees of freedom (samples minus model parameters) determine the inverse of the statistical complexity of the model?*


## 9. Conclusions

While NSM has advanced the modeling of uncertainty within complex systems, there remain many open problems worthy of investigation. In this paper, issues arising from the use of the parameter *q* as a focal point for modeling complex systems are examined. These issues are framed in terms of a set of open problems, including the following:Should the generalized exponential function, originally proposed by (Borges, 2004) [26], be applied to the survival function rather than the probability density functions?Can the difference between generalized entropy and BGS entropy be explained in terms of the degrees of freedom and its inverse, the nonlinear statistical coupling?For NSM to be a complete physical theory, a clear physical interpretation of *q* is required. We must determine whether the number of independent random variables sharing the same state is the appropriate interpretation of *q*.We must define the *q*-product using the properties of the multivariate distributions of *q*-statistics.The *q*-Fourier transform does not seem to model the physical image of a heavy-tailed signal. For example, the Cauchy distribution is transformed into a delta function, which could not be used for real-world signal processing. Can a generalization of the Fourier transform be defined that utilizes the complementary properties of the compact-support and heavy-tailed domains?The normalization of the coupled entropy differs from both the normalized and unnormalized Tsallis entropy. Is there a criterion that would clarify a preference between these three normalizations of the generalized entropy for complex systems?We must define a measure of statistical complexity.

The author has proposed that the nonlinear statistical coupling, which is equal to the shape parameter and the inverse of the degrees of freedom, is a measure of statistical complexity. It is left to the reader to examine this set of open problems, determine satisfactory solutions, and consider whether a reframing of nonextensive statistical mechanics leads to a focus on the fundamental properties of complex systems.

## Figures and Tables

**Figure 1 entropy-26-00118-f001:**
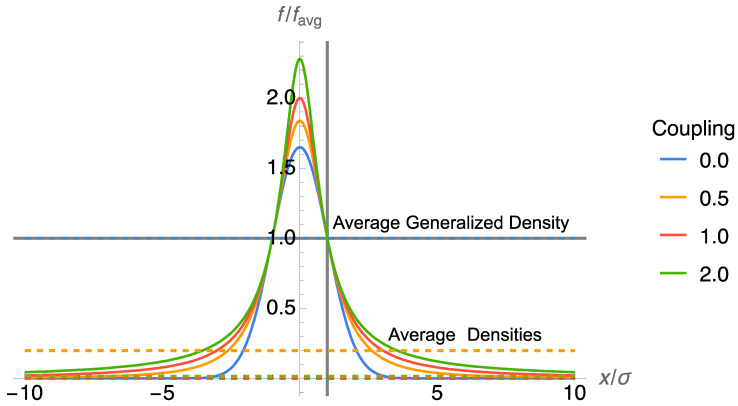
Contrast between the average density and average generalized density.

**Table 1 entropy-26-00118-t001:** Applications of nonextensive statistical mechanics. A variety of complex systems, such as atomic gases, space plasma velocities, financial volatility, cellular mobility, wavelets, and heat baths, can be modeled using NSM. In each case, the mapping between the physical property and the parameter *q* requires numerical constants that diminish the ability of *q*-statistics to describe the physical phenomena.

Applications	Physical Property	Relation to *q*
Entropy of Hydrogen Atoms [17]	M, number of atoms	q=1+1M
Space Plasma Velocities [18,19]	κ=ν, spectral index	q=1+1κ
Volatility of Financial Markets [7,20]	ν, nonlinear Fokker-Plank	q=2−ν
Hydra-Cell Velocity [21,22]	ν, nonlinear Kramers Equ	q=2−ν
Wavelet Analysis [23]	*i*, wavelet scale index	q=1−2i
Heat Bath Thermodynamics [24,25]	*n*, degrees of freedom, $n=\frac{d\;N}{2}$, *d* dimensions, *N* particles	$q=\frac{n}{n-1}$
Superstatistic Fluctuations [5]	*n*, Chi-square deg. of freedom	$q=1+\frac{2}{n+1}$

**Table 2 entropy-26-00118-t002:** Description of the *q*-Gaussian domains and their associated *q*-FT image. The *q*-FT transforms have functions such as the q-FT to images with slower decaying tails. So, for example, the last row is the domain of distributions with an undefined mean, which has an image with a divergent integral.

*q*-Gaussian Domain			*q*-FT Image		
Description	Shape, *κ*	*q*	Description	Shape	*q*
Finite Mean Finite Var.	0, 13	1, 32	Finite Mean Finite Var.	0, 12	1, 53
Finite Mean Finite Var.	13, 12	32, 53	Finite Mean Div. Variance	12 ,1	53, 2
Finite Mean Div. Variance	12 , 1	53, 2	Undefined Mean Div. Variance	[1, ∞)	[2, 3)
Undefined Mean Div. Variance	[1, ∞)	[2, 3)	Div. Integral	[−∞,−1)	[3, ∞)

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
