# Peer review of "Open Problems within Nonextensive Statistical Mechanics"

_entropy, 2024, doi:10.3390/e26020118_

Round 1

Reviewer 1 Report

Comments and Suggestions for Authors

In the work “Open problems within Nonextensive Statistical Mechanics” by Kenric P. Nelson., the Authors review the current status of the Tsallis statistics and its applications. After the Authors, the review starts by grounding q-statistics within scale-shape distributions and then frames a series of open problems for investigation. The open problems include using the degree of freedom to quantify the difference between entropy and its generalization; clarifying the physical interpretation of the parameter q; improving the definition of the generalized product using multidimensional analysis; defining a generalized Fourier transform applicable to signal processing applications; and re-examination of the normalization of nonextensive entropy. The review concludes with a proposal that the shape parameter is a candidate for defining the statistical complexity of a system.

The style of writing is unorthodox but the manuscript reads very well. It appears to be free of major errors and deals with timely and important aspect of Tsallis statistics. Hence, I would like to recommend this study for publication. I have only two minor comments:

1.     This study reads almost as a review article, which is a positive aspect in my opinion. However, at the same time the text is bit too general in some places. My main critique relates to the part at the very beginning of the manuscript, where the mentioned applications of Tsallis statistics are described. As mentioned, this description is very general and it points out to the general references. To make this part of the text more instructive to the readers I suggest adding several specific and recent application examples of the Tsallis statistics e.g. in terms of statistics of molecular systems (refer to Physica A 518 (2019) 1-12), entropy of a black holes (refer to European Physical Journal C 82 (2022) 713) or an alternative measure of financial data volatility (refer to Risks 9 (2021) 89).

2.     The Authors often mention that Tsallis statistics is well suited to deal with the heavy tailed distribution. This kind of distributions are present in case of financial data, contrasting with the normality of many current econometric models. This is especially visible in terms of the aforementioned volatility of financial data (see Entropy 25 (2023) 823). In this context, it has been shown that Shannon entropy deals much better with the heavy tailed distributions than standard deviation, the conventional measure of volatility (please refer again to Entropy 25 (2023) 823). I think that it is worth to mention this field of applications where the Tsallis entropy is compared with the Shannon entropy. However, it is instructive to do so noting that even Shannon entropy provides substantial improvement over standard deviation. To this end, what is the opinion of Author on the potential of using Tsallis entropy for financial data instead of Shannon entropy? In this respect, what are the improvements provided by the Tsallis entropy comparing to the Shannon approach?

Author Response

Thank you for your recommendation that this review article be published in Entropy.

1) I added a paragraph to the introduction reviewing NSM applications in more detail. Included is a table relating the parameter q to a physical property. It is pointed out that the appearance of constants in each of these expressions diminishes the ability of q to be a description of the properties of complex systems.

2) The application of Tsallis entropy to financial analysis is added in both the introduction and section 7. While the references discuss the merits of generalized entropies, the focus here is on whether a more direct relationship between the physical phenomena and the definition of generalized entropy would improve the interpretation of the results.

Reviewer 2 Report

Comments and Suggestions for Authors

The paper "Open problems within Nonextensive Statistical Mechanics" presents a brief analysis to celebrate the 80th birthday of Constantino Tsallis, by exploring open problems that can inspire future research in the field. The idea behind this review is interesting and offers a roadmap for future exploration and advancement in non-extensive statistical mechanics. However, this work should be considered a perspective rather than a review article. In addition, although the paper deals with an exciting subject, the presentation of this work needs to be improved since there are missing or poorly motivated definitions. Please see the comments below.

1) Does the author consider that the property exp_q(x) exp_q(-x) not equal to 1 is not one of the major problems associated with Tsallis statistics? Does the author consider this to be an open problem?

2) The author does not explicitly address challenges related to the field of machine learning using NSM. Nevertheless, it raises the question of whether the integration of machine learning techniques with NSM could be explored in future research. This aspect requires further discussion.

3) The authors are encouraged to delve into a meticulous review of the manuscript, as some typos and instances of informal language are woven into the text. For instance, consider replacing 'Boltzmann-Gibb statistics' with 'Boltzmann-Gibbs statistics' on the second page.

4) Define the Lagrange multiplier (beta) in Equation 1.

5) Maintain the linearity of the writing. At times, the parameters q and beta are in bold, while in other instances, they are not.

6) Please use the correct capitalization in "Problems" in the title.

Comments on the Quality of English Language

A proofreading is needed.

Author Response

Thank you for the suggestions on improving the paper.

While I have referred to nonlinear statistical coupling as a perspective on nonextensive statistical coupling in papers such as "On the average uncertainty for systems with nonlinear coupling", Physica A 468 (2017), in this paper I am seeking to provide an objective review of problems within NSM. That said, the paper does not seek to be a comprehensive review of the field.

1) After equation 14, a comment describing the distinction between the reciprocal of the generalized exponential and the negative of its argument is added.

2) Thank you for the suggestion to include applications to machine learning. I have published a paper training a variational autoencoder with the coupled entropy. This and a related paper using the q-entropy is added as a comment to Problem 6. A determination of whether the distinction in normalization impacts the performance is suggested.

3)  A careful proofread of the document including a review with Grammarly was completed. All of the changes are highlighted in yellow.

4) After equations 1 and 2 the symbol beta is labeled as the Lagrange multiplier. Definitions for the hypergeometric and beta functions are added to equations 8 and 9.

5) Microsoft Word is automatically converting some symbols to bold. I was careful to make sure the pdf wasn't saved in this form. Bold symbols are used for the vectors p and x only.

6) The capitalization of the title is correct.

Again, thank you for the suggestions.